# Learning to Combat Compounding-Error in Model-Based Reinforcement Learning

## Abstract

Despite its potential to improve sample complexity versus model-free approaches, model-based reinforcement learning can fail catastrophically if the model is inaccurate. An algorithm should ideally be able to trust an imperfect model over a reasonably long planning horizon, and only rely on model-free updates when the model errors get infeasibly large. In this paper, we investigate techniques for choosing the planning horizon on a state-dependent basis, where a state's planning horizon is determined by the maximum cumulative model error around that state. We demonstrate that these state-dependent model errors can be learned with Temporal Difference methods, based on a novel approach of temporally decomposing the cumulative model errors. Experimental results show that the proposed method can successfully adapt the planning horizon to account for state-dependent model accuracy, significantly improving the efficiency of policy learning compared to model-based and model-free baselines.

## 1 Introduction

Model-free reinforcement learning (RL) aims to learn an effective behavior policy directly from interaction with a black-box environment. This approach has recently achieved great success, particularly in game playing (Mnih et al., 2015; Moravčík et al., 2017; Silver et al., 2016; 2017). Unfortunately, model-free RL techniques are hampered by poor sample efficiency, which makes their deployment infeasible whenever data collection is expensive. A key challenge remains to improve the sample efficiency of general purpose RL methods.

By contrast, model-based RL attempts to learn a model of an environment from direct experience collected during training. A learned model can be either directly combined with a planning algorithm (Hafner et al., 2018; Sutton, 1990), or applied to improve the target values for model-free RL (Buckman et al., 2018; Feinberg et al., 2018). Model-based RL is often thought to be more sample efficient than model-free approaches (Sutton & Barto, 2018). Recent theoretical work confirms this intuition by showing that there exist environments where model-based approaches can be exponentially more sample efficient than any model-free approach (Sun et al., 2018).

The performance of model-based RL heavily relies on the quality of the model a learning agent can acquire. When an accurate model is given or can be learned with relatively little experience, model-based RL can be significantly more data efficient than model-free approaches. However, in noisy and complex environments, learning an accurate model can be a challenge. In such cases, model errors can compound and render the model useless for planning, which can lead to catastrophic failure of model-based RL. Although having an accurate model in a complex environment can be unrealistic, it is sometimes possible to obtain a model that is accurate in local subsets of the state space. For example in robotic control tasks, local-motion dynamics that do not consider external environment interaction can be much easier to model than the dynamics of interaction with other objects. In such cases, even if a pure model-based approach would fail, one might still expect to gain advantage over model-based approaches by exploiting the accurate parts of the model.

One potential advantage of model-based RL is that a longer planning horizon can be considered by rolling out the model for multiple steps. Ideally, with an imperfect model, one would like a principled approach for adapting the planning horizon at different states, in order to overcome the compounding error problem of model-based RL. When the model has large error around some states, the learning agent should trust the model less by using a small planning horizon.

On the other hand, for states where the model is near optimal, a large planning horizon should be adopted. To implement this idea, we provide a few key observations that are essential for the methods we propose: First, we characterize the error in a multi-step learning target under an approximate model as a multi-step discounted cumulative model error. Second, we show how the cumulative model error for different planning horizons can be learned based on TD-learning. Finally, we introduce *Adaptive Model-based Value Expansion* (AdaMVE), an extension to *Model-based Value Expansion* (MVE) Feinberg et al. (2018) that adaptively selects planning horizons for each state, based on the learned accumulative model errors. To illustrate, Fig. 1 provides an example of how AdaMVE works in a FourRoom gridworld maze with an imperfect model. This example shows that AdaMVE can successfully adapt the plan-

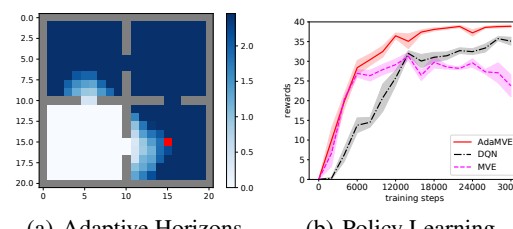

(a) Adaptive Horizons  (b) Policy Learning

Figure 1: Illustration of adaptive planning horizon in FourRoom with an imperfect model. The model is perfect in three rooms while totally wrong in the left bottom room. Non-adaptive MVE diverges due to the large model errors (right). In contrast, AdaMVE is able to adapt the planning horizon at different state (see (a), darker color means longer planning horizon), outperforming both the model-based and model-free baselines.

ning horizon for different subsets of the state space, and significantly outperforms both MVE and model-free baselines. We evaluate our method on both gridworld mazes with different imperfect models as well as continuous control tasks. The experimental results suggest that the adaptive horizon algorithm can significantly alleviate the compounding error problem in model-based RL, while exploiting its advantage in terms of sample efficiency.

## 2 BACKGROUND

Consider a Markov Decision Process (MDP) Sutton & Barto (2018) $\mathcal{M} = (\mathcal{S}, \mathcal{A}, P, R, \gamma)$ where $\mathcal{S}$ is the state space, $\mathcal{A}$ is the action space, $P(\cdot|s, a)$ is the transition probability distribution function, $R(s, a)$ is the reward function and $\gamma$ is the discount factor. We also assume that each state is represented by a feature vector $s \in \mathbb{R}^d$. The goal is to find a policy $\pi(\cdot|s)$ that maximizes the cumulative discounted reward starting from any state $s \in \mathcal{S}$. Let $P^\pi(\cdot|s)$ denote the induced transition distribution for policy $\pi$. For later convenience, we also introduce the notion of multi-step transition distributions as $P_t^\pi$, where $P_t^\pi(\cdot|s)$ denotes the distribution over the state space after rolling out $P^\pi$ for $t$ steps starting from state $s$. For example, $P_0^\pi(\cdot|s)$ is the Dirac delta function at $s$ and $P_1^\pi(\cdot|s) = P^\pi(\cdot|s)$. We use $R^\pi(s)$ to denote the expected reward at state $s$ when following policy $\pi$, i.e. $R^\pi(s) = \mathbb{E}_{a \sim \pi(\cdot|s)} [R(s, a)]$. The state value function is defined by

$$V^\pi(s) = \sum_{t=0}^{\infty} \gamma^t \mathbb{E}_{s_t \sim P_t^\pi(s)} [R^\pi(s_t)] .$$

The action-value function (a.k.a. Q-function) can be written as

$$Q^\pi(s, a) = R(s, a) + \gamma \mathbb{E}_{s' \sim P(\cdot|s,a)} [V^\pi(s')] .$$

The optimal policy is define as the policy $\pi$ that maximizes $V^\pi(s)$ at all states $s \in \mathcal{S}$. Such policy always exists (Sutton & Barto, 2018).

*Model-based* reinforcement learning approaches explicitly make use of a dynamics model $\hat{P} \approx P$ of the environment to compute the optimal policy, while *model-free* approaches learn the optimal policy without explicitly modeling $P$ (e.g. directly learning the action-value functions from samples). Throughout the paper we assume that the reward function $R$ is known to the agent, hence a "model" refers to an estimated transition dynamics $P$.

### 2.1 MULTI-STEP MODEL-BASED VALUE

One potential advantage of learning a model is to compute a multi-step target value by iteratively rolling out the model, that is, to take the predicted state of the model and feed it in again as the state input, projecting to a sample state two time steps later, and so on. Formally, for any policy $\pi$, given a

planning horizon $H$, a reference (target) value function $\bar{V}$, and an approximate (e.g. learned) model $\hat{P}$, the *$H$-step model-based value* is defined as

$$\hat{V}_{\hat{P},H}^{\pi}(s) = \sum_{t=0}^{H-1} \gamma^t \mathbb{E}_{s_t \sim \hat{P}_t^{\pi}(s)}\left[R^{\pi}(s_t)\right] + \gamma^H \mathbb{E}_{s_H \sim \hat{P}_H^{\pi}(s)}[\bar{V}(s_H)]. \tag{1}$$

This value can be integrated with model-free methods in different ways based on the policy $\pi$. For example in AlphaGo Zero, the $H$-step optimal lookahead policy $\pi_H^* = \operatorname{argmax}_\pi \hat{V}_{\hat{P},H}^{\pi}(s)$ combined with a proper exploration strategy is used as the behavior policy of the learning agent, where $\pi_H^*$ is approximated by Monte Carlo Tree Search (Silver et al., 2017).

Model-based value expansion (MVE) is another example of utilizing objective (1) (Buckman et al., 2018; Feinberg et al., 2018). MVE applies the learning agent's current policy as the rollout policy to obtain $\hat{V}_{\hat{P},H}^{\pi}(s)$, which is used as the update target value for TD Learning. For example in Q-learning (Watkins & Dayan, 1992), given a sampled transition $(s, a, r, s')$ and a target Q-value function $\bar{Q}(s, a)$, one can replace the target value $\bar{V}(s') = \max_a \bar{Q}(s', a)$ with the multi-step estimate $\hat{V}_{\hat{P},H}^{\pi}(s')$ or a mixture of such estimates with different values of $H$. The rollout policy $\pi$ can be greedy with respect to $\bar{Q}$. Note that when $H = 0$, there is no rollout hence $\hat{V}_{\hat{P},0}^{\pi}(s') = \bar{V}(s')$, which recovers the model-free update target.

When the model is perfect, MVE can reduce the biases of the targets, leading to improved performance over model-free methods (Feinberg et al., 2018). However, the major limitation of MVE is that the rollout horizon $H$ needs to be tuned in a task-specific manner: in a complex environment where the model is difficult to learn, a smaller rollout horizon usually performs better than a larger one. To overcome this drawback, Buckman et al. propose stochastic ensemble value expansion (STEVE), which applies stochastic ensembles both over multiple models and rollout horizons to choose the best $H$ dynamically (Buckman et al., 2018).

## 2.2 WASSERSTEIN DISTANCE

The Wasserstein distance is a distance metric between two distributions. Its Kantorovich-Rubinstein dual form is defined as follows (Villani, 2008):

$$W(p, q) = \sup_{\|g\|_L \le 1} \mathbb{E}_{z \sim p}\left[g(z)\right] - \mathbb{E}_{z \sim q}\left[g(z)\right] \tag{2}$$

where $\|g\|_L$ is the Lipschitz constant of function $g$. If both $p$ and $q$ are Dirac delta functions (i.e. deterministic) denoted by $\delta_{z_p}$ and $\delta_{z_q}$ respectively, $W(p, q) = \|z_p - z_q\|_2$ is just the Euclidean distance. If only one of the distributions is deterministic, e.g. $q = \delta_{z_q}$, then $W(p, q) = \mathbb{E}_{z \sim p}\left[\|z - z_q\|_2\right]$. This can be checked directly from the primal form or observing that $g(z) = \|z - z_q\|_2$ achieves the superimum in the dual form (Villani, 2008).

## 3 LEARNING MULTI-STEP MODEL ERROR

To exploit the advantage of multi-step value estimation while alleviating the negative impact of using an imperfect model, we propose to adapt planning horizons $H$ such that the $H$-step expanded value using the approximate model is close to the one obtained using the true model. Specifically, for policy $\pi$, consider the *$h$-step model-based value error* for an approximate model $\hat{P}$ defined by

$$\mathcal{E}(h|\pi, s, \bar{V}, \hat{P}) = \left|\hat{V}_{\hat{P},h}^{\pi}(s) - \hat{V}_{P,h}^{\pi}(s)\right|. \tag{3}$$

We aim to select an appropriate planning horizon for state $s$ based on this error. Since the accuracy of an approximate model could vary in different subspace of $\mathcal{S}$, selecting planning horizons in a state dependent way is particularly desirable in practice.

Given a state $s$, exactly computing error (3) is unfeasible as we cannot directly compute the multi-step model-based value due to the inaccessibility of the true model $P$. To overcome this problem, we propose a practical algorithm to learn the value expansion error for each state approximately, based on the observation that the value expansion error defined in (3) can be characterized by the discounted accumulative model error. The following theorem states such connection.

**Theorem 1.** *Given any policy $\pi$, an approximate model $\hat{P}$, and a reference value function $\bar{V}$, for planning horizon $H$ we have*

$$\mathcal{E}(H|\pi, s, \bar{V}, \hat{P}) = \left| \hat{V}^{\pi}_{\hat{P},H}(s) - \hat{V}^{\pi}_{P,H}(s) \right| \leq K \cdot \sum_{t=0}^{H-1} \gamma^{t+1} \mathbb{E}_{s_t \sim P^{\pi}_t(s)} \left[ W^{\pi}(s_t) \right] , \tag{4}$$

*where $W^{\pi}(s) = \mathbb{E}_{a \sim \pi(\cdot|s)} \left[ W(s, a) \right]$ with $W(s, a) = W(P(\cdot|s, a), \hat{P}(\cdot|s, a))$ being the Wasserstein distance and $K = \sup_h \left\| \hat{V}^{\pi}_{\hat{P}, h} \right\|_L$ is the maximum Lipschitzness of the estimated value function over all possible horizons.*

The $H$-step discounted cumulative model error (RHS of (4)) can be viewed as a finite-horizon RL objective. We can define a new MDP to learn this error, $\mathcal{M}_{H,\hat{P}} = (\mathcal{S}, \mathcal{A}, P, W, \gamma)$, where the state and action space, the transition function, and the discount factor remain unchanged from the original problem, the *W-reward function* $W(s, a) = W(P(\cdot|s, a), \hat{P}(\cdot|s, a))$ defines the "reward" of the MDP. With $\mathcal{M}_{H,\hat{P}}$, we define a new state-value function named *h-step state model error function*, which measures the expected cumulative model error if the agent starts in state $s$ and follows some policy $\pi$,

$$\hat{\mathcal{E}}^{\pi}(s, h) = \begin{cases} 0 & h = 0 \\ \sum_{t=0}^{h-1} \gamma^t \mathbb{E}_{s_t \sim P^{\pi}_t(s)} \left[ W^{\pi}(s_t) \right] & h > 0 \end{cases} \tag{5}$$

According to (4), $\hat{\mathcal{E}}^{\pi}(s, h)$ is an upper bound of the $h$-step model-based value error (3) up to a constant. Similarly, the *h-step action model error* is defined by

$$\hat{\mathcal{E}}^{\pi}(s, a, h) = \begin{cases} 0 & h = 0 \\ W(s, a) + \gamma \mathbb{E}_{s' \sim P(\cdot|s, a)} \left[ \hat{\mathcal{E}}^{\pi}(s', h - 1) \right] & h > 0 \end{cases} \tag{6}$$

Note that $\hat{\mathcal{E}}^{\pi}(s, h) = \mathbb{E}_{a \sim \pi} \left[ \hat{\mathcal{E}}^{\pi}(s, h, a) \right]$ by the definition. Learning the $h$-step model-based value error under some policy $\pi$ now becomes a traditional policy evaluation problem in RL with a different reward function. In this paper we use mode-free methods to learn the model error function.

## 3.1 Learning $h$-step Cumulative Model Error

We now introduce a principled method to learn the cumulative model error based on finite horizon Bellman updates. Since $\mathcal{M}_{H,\hat{P}}$ only differs from the original MDP in the reward function, we can directly learn the $h$-step model error using the transition data sampled from a relay buffer. As discussed in Section 2.2, when either the ground-truth or the approximate transition is deterministic, the W-reward is just the expected Euclidean norm between the real and predicted next state. For stochastic transitions, W-reward can be approximated by learning the $g$ function in (2) as suggested in the Wasserstein GAN (Arjovsky et al., 2017). Therefore, given a sampled transition data $(s_t, a_t, r_t, s_{t+1})$, we can directly compute the W-reward for learning. In addition, since the policy $\pi$ used for computing mode-based value is non-stationary during learning, which may or may not be available at the time of evaluating the value expansion errors, we measure the value expansion error using a reference policy $\bar{\pi}$. The choice of $\bar{\pi}$ will be discussed later.

We use TD-learning to train the model error function. In particular, given a data sample $(s, a, r, s')$, $\hat{\mathcal{E}}$ is updated by minimizing the one step Bellman error

$$\min_{\hat{\mathcal{E}}} \frac{1}{2} \left\{ W(s, a) + \gamma \bar{\mathcal{E}}(s', a', h - 1) - \hat{\mathcal{E}}(s, a, h) \right\}^2 \tag{7}$$

where $a'$ is selected using the policy $\bar{\pi}$, and $\bar{\mathcal{E}}$ is the target model error value function. It is important to note that this learning process can be combined with any model-based algorithm where a replay buffer is used. Also, our method does not introduce additional sample complexity, since the data used to train (7) can be sampled from the replay buffer.

**Choice of the Reference Policy**. Since the policy used for value expansion is changing during the learning process and the cumulative model error is policy dependent, it is expensive to retrain the model error for the current policy at every step. Thus, we choose the reference policy $\bar{\pi}$ to "prepare for the future". There are several possibilities but we consider three that allow model error to be efficiently learned with samples from the replay buffer.

1. The *conservative reference policy* $\operatorname{argmax}_\pi \mathcal{E}(h|\pi, s, \bar{V}, \hat{P})$ targets the maximum model error. When using the learned model error to decide a proper planning horizon, this policy allows us to consider the worst case model error, making the selected horizon more robust in practice. To learn the model error under this policy, we can use $a' = \operatorname{argmax}_a \hat{\mathcal{E}}(s', a, h-1)$ when doing the update (7). This can be viewed as an extension of Q-learning (Watkins & Dayan, 1992) to learn the maximum model error.

2. The *greedy reference policy* selects $\operatorname{argmax}_a \bar{Q}(s, a)$, where $\bar{Q}$ is the current target Q-value function of the learning agent. This policy tries to measure the model error under the learning agent's current behavior.

3. The *replay buffer reference policy* selects an action which occurred in the replay buffer at $s$. This policy can be considered as a mixture policy of previous agent's behaviors. As we only care about the "value function" $\hat{\mathcal{E}}^{\bar{\pi}}(s, h)$ and every sampled transition $(s, a, r, s')$ can be viewed as "on-policy" under the replay buffer policy, we can directly estimate $\hat{\mathcal{E}}^{\bar{\pi}}(s, h)$ by on-policy TD learning: $\min_{\hat{\mathcal{E}}} \frac{1}{2}\{W(s, a) + \gamma \bar{\mathcal{E}}(s', h-1) - \hat{\mathcal{E}}(s, h)\}^2$.

## 3.2 ADAPTIVE PLANNING HORIZON USING MODEL ERROR

In this section, we introduce the *Adaptive Model-based Value Expansion* (AdaMVE) algorithm, as an example of using a learned multi-step model error function to adapt planning horizon for different states. Instead of applying a fixed horizon tuned for different environment when computing (1) as in MVE, AdaMVE attempts to adapt the rollout horizon for different states according to a model error function learned as described in the previous section. We suppose that the learner's behavior policy $\pi$ is determined by Q-values. For a discrete domain, $\pi = \operatorname{argmax} Q(s, a)$. For a continuous domain, $\pi$ is trained to approximate the greedy policy over $Q(s, a)$ as in DDPG (Lillicrap et al., 2015).

In AdaMVE, we aim to find a proper planning horizon $H(s) \in [0, H_{\max}]$ for any state $s \in \mathcal{S}$ [1]. For state $s \in \mathcal{S}$, we use the learned model error function $\hat{\mathcal{E}}$ to produce the model error $\hat{\mathcal{E}}(s, h)$ for all rollout horizons $h \in [0, H_{\max}]$. Although $\hat{\mathcal{E}}$ can be viewed as a good proxy for $\mathcal{E}$ in (4), it is still difficult to directly find an appropriate planning horizon by setting a hard threshold, due to the unknown Lipschitz constant $K$ in Theorem 1. To resolve this issue, instead of setting a hard threshold to get a maximum rollout horizon $H(s)$, we use a soft weighted combination over all horizons $h \leq H_{\max}$. For horizon $h$ with higher model error $\hat{\mathcal{E}}(s, h)$ we set a smaller weight $\omega_h$. More specifically, we define the weights according to a "softmax policy"

$$\omega(h|s) \propto \exp\left\{-\hat{\mathcal{E}}(s, h)/\tau\right\} \tag{8}$$

where $\tau$ is a temperature parameter. AdaMVE uses this policy as its weighting function to mix the expansion values (1) of different rollout horizons,

$$\tilde{V}^\pi_{\hat{P}, H_{\max}}(s) = \sum_{h=0}^{H_{\max}} \omega(h|s) \hat{V}^\pi_{\hat{P}, h}(s). \tag{9}$$

To give a scalar measure of the "planning horizon" when using soft combination, for each state $s$ we define its **weighted average horizon** as $\bar{H}(s) = \sum_{h=1}^{H_{\max}} \omega(h|s) * h$, which is the expected rollout horizon under a "softmax policy". When the model error is zero everywhere, $\omega(h|s) = 1/(H_{\max}+1)$ and $\tilde{V}^\pi_{\hat{P}, H_{\max}}(s)$ is the average value estimates of all horizon. The average planing horizon is $\bar{H}(s) = H_{\max}/2$. When the model error is infinitely large everywhere, $\omega(0|s) = 1$ and $\omega(h|s) = 0$ for all $h > 0$ thus no rollout is being considered. The average planing horizon is $\bar{H}(s) = 0$.

The combined value estimate (9) is used as the learning target to update $Q(s, a)$. Specifically, at each training step, AdaMVE samples a batch of data $(s, a, r, s')$ from a replay buffer $\mathcal{B}$. For each $s'$ in the batch, an on-policy $H_{\max}$-step rollout is computed using the model and the learning agent's current policy $\pi$. For each data $(s, a, r, s')$, we update $Q$ by minimizing $\frac{1}{2}\{r + \gamma \tilde{V}^\pi_{\hat{P}, H_{\max}}(s') - Q(s, a)\}^2$, where a target Q-value is used to compute the value at the end state of each rollout. Pseudocode of AdaMVE is provided in the Appendix.

---

[1] $H_{\max}$ is set by computational consideration in practice. Our proposed method is valid for any large $H_{\max}$.

## 4 RELATED WORK

Previous work in model-based reinforcement learning can be divided in two categories: using the model for planning in low-dimensional state spaces, and combining the benefits of model-based and model-free approaches. For the first category, Gal et al. (Gal et al., 2016) combine the PILCO algorithm (Deisenroth & Rasmussen, 2011) with a neural dynamic model. Hafner et al. (Hafner et al., 2018) propose to learn a latent dynamic model and choose actions through online planning with the latent model. For the second category, Weber et al. use imaginary rollouts generated by the dynamic model for policy learning (Racanière et al., 2017). Gu et al. propose to augment imaginary rollout data to the experience replay buffer and show that this can accelerate model-free learning (Gu et al., 2016). In this paper, we use MVE (Feinberg et al., 2018) as the baseline algorithm to show the effectiveness of our adaptive planning horizon algorithm. But it is important to note that our method can be combined with any of the model-based methods discussed above.

The compounding error phenomenon of model-based RL is previously discussed in (Asadi et al., 2018; Jiang et al., 2015; Talvitie, 2017; Wang et al., 2019). Surprisingly, relatively little work has been done to solve this problem. Buckman et al. propose STEVE, which uses stochastic ensemble of models and planning horizons to relax the error caused by using only one model with a fixed planning horizon (Buckman et al., 2018). The ensemble with lowest variance is used as the learning target. In comparison, our proposed method directly handles the compounding error by adaptively selecting planning horizons based on a learned model error function. In addition to using a different design idea, our approach has both the model and the model error as a single function, which is far less computationally expensive than STEVE, that learns multiple models in an ensemble. It is also worth noting that the model in our approach can be either hand designed, pretrained from another task, or learned online.

## 5 EXPERIMENTS

We conduct experiments on both gridworld and continuous control environments. For the gridworld environment, we implement our adaptive value expansion (AdaMVE) based on DQN Mnih et al. (2015) and compare to the vanilla DQN and its non-adaptive value expansion variant (MVE). For continuous control, we implement AdaMVE with DDPG Lillicrap et al. (2015) and compare to the vanilla DDPG and MVE. We perform a single update to the policy for all methods in comparison at each environmental step.

### 5.1 EXPERIMENTS ON GRIDWORLD

**Visualizing the Adaptive Horizon.** We first evaluate the learned adaptive planning horizon by visualizing in a gridworld with predefined imperfect model. If our method works correctly, we should observe a large horizon at states where the model is accurate, but small horizon for those states where the model deviates a lot from the true environment dynamics.

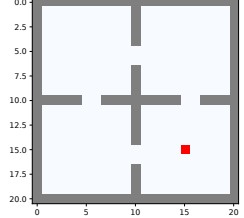

Figure 2: FourRoom Env

We use a FourRoom gridworld maze (Fig. 2). Each room has $9 \times 9$ cells. The agent's objective is to find the goal position (in red) starting from a random initial position. There are five actions: left, right, up, down and stay. The maximum length of each episode is 50. After each episode the agent restarts in a random position. The reward is 1 when hitting the goal and 0 in the other positions. A state is represented by the $(x, y)$ coordinate. We evaluate the adaptive planning horizon using the following models:

- *Oracle model*. The transition function behaves exactly the same as the true environment.
- *3Room model*. The transition function is true in three rooms, but completely wrong in the left bottom room. For a given state and action in this room, the model simply produces a randomly sampled next state from all possible positions.
- *NoWall model*. This model ignores the existence of the wall. For example, given a state at one side of a wall, if the agent takes the action towards the wall, this model will predict the position overlapping with the wall as the next state, but in the true environment the agent will just stop at the current position.

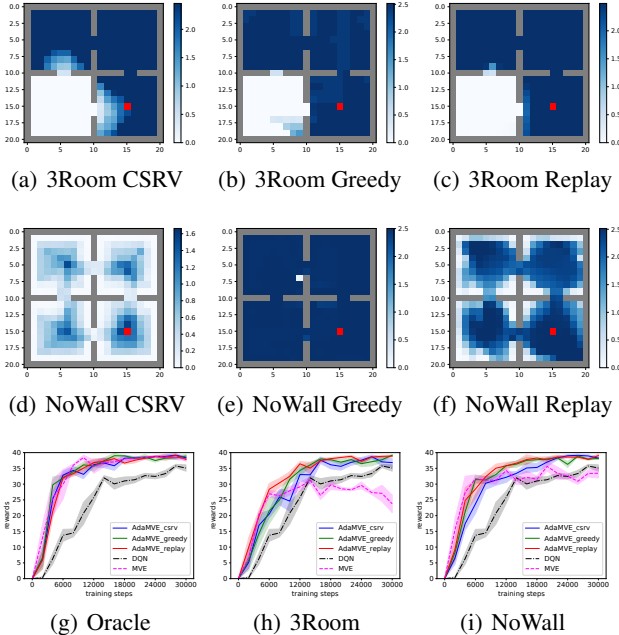

Figure 3: Experiment results on FourRoom. (a)-(f) Visualization of learned planning horizon for all states. For each state, the *average horizon* $\bar{H}(s)$ weighted by (8) is presented. We use $H_{\max} = 5$ thus $\bar{H}(s) <= H_{\max}/2 = 2.5$ from definition. Our method can successfully adapt the planning horizon when the model is imperfect. (g)-(i) Policy learning performance with different models. The shaded area shows the standard error. Results clearly show that AdaMVE significantly outperforms MVE when the model is imperfect (no wall model and 3room model).

We evaluate the three reference policies (conservative, greedy and replay buffer) discussed in Section 3.1, denoted by *csrv*, *greedy* and *replay* respectively. The results are visualized in Fig. 3 (a)-(f). For each state (position), we show its *weighted average horizon* $\bar{H}(s)$, as defined in Section 3.2. The results clearly show the effectiveness of the proposed method in finding an appropriate planning horizon for different parts of the state space. For example, in the 3room model, our method can adopt zero planning horizon for states in the left bottom room where the model has large error.

**Policy Learning Performance in FourRoom.** We compare AdaMVE with MVE and DQN using the three models described above. Results are presented in Fig. 3 (g)-(i). Each data point is averaged over 5 runs. Each run is evaluated after every 2000 environmental steps by computing the mean total episode reward across 10 episodes. When the oracle model is available, the model error is zero everywhere, hence AdaMVE performs exactly the same with MVE. Both algorithms outperforms DQN, which confirms that the model-based value expansion targets can lead to improved performance. However, when the model is noisy, MVE diverges due to model errors. In contrast, AdaMVE still converges, and does so significantly faster than DQN. This is because AdaMVE can adapt the rollout horizon for states where the model has large error as shown in the visualization. For example, when using the 3room model, AdaMVE only performs the model-free updates for states in the bottom left room. In contrast, at these states MVE still trusts the multi-step value expansion target, which has large error that causes diverge.

## 5.2 CONTINUOUS CONTROL

We experiment on continuous control tasks to further verify the benefit of adaptive rollout horizons. We first use Mujoco (Todorov et al., 2012) to create 3D mazes and learn to control a PointMass agent to navigate to a goal area in the maze, starting from a random location, as shown in Fig 4(a-c). We also test on two Mujoco control problems in OpenAI Gym (Brockman et al., 2016): HalfCheetah and Swimmer. All results are based on 5 different runs. Each data point in the plots is evaluated by 200 test episodes for PointMass Navigation, and 50 test episodes for Mujoco control problems.

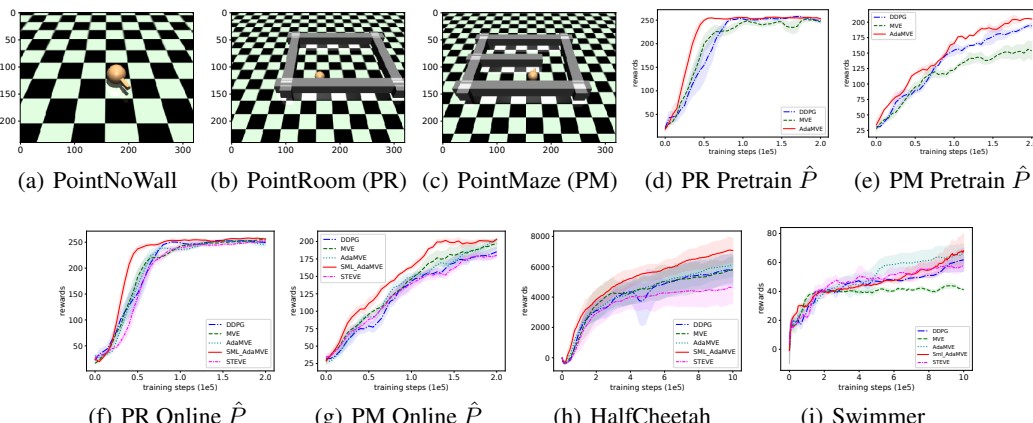

(a) PointNoWall    (b) PointRoom (PR)    (c) PointMaze (PM)    (d) PR Pretrain $\hat{P}$    (e) PM Pretrain $\hat{P}$

(f) PR Online $\hat{P}$    (g) PM Online $\hat{P}$    (h) HalfCheetah    (i) Swimmer

Figure 4: Results of continuous control. (a)-(c) PointMass Navigation example environments. (d)-(e) PointMass Navigation policy learning performance using pretrained model. (f)-(g) PointMass Navigation policy learning performance using online learned model. (h)-(i) Policy learning performance on HalfCheetah and Swimmer. AdaMVE outperforms both DDPG and MVE with pretrained model. With online learned model, vanilla AdaMVE does not improve performance over the baselines. SML_AdaMVE outperforms all the baseline algorithms with an online learned model.

**Results with Pretrained Model.** We first evaluate the adaptive method using a pretrained model on the PointMass Navigation problem. We pretrain a model by executing a uniform-random policy in PointNoWall (Fig 4 (a)). This pretrained model is then used as an imperfect model in PointRoom (PR, Fig 4 (b)) and PointMaze (PM, Fig 4 (c)) without further training. This model is supposed to be good at modeling local motions while bad at modeling interactions with the wall. For AdaMVE, we use the *replay* reference policy and $H_{\max} = 5$. MVE applies a fixed rollout horizon 5. As shown in Fig 4 (d)-(e), AdaMVE outperforms both MVE and DDPG, which further justifies the benefits of selecting rollout horizons in an adaptive way.

**Results with Online Learned Model.** We next conduct experiments with an online learned model. The model is updated by one gradient step at each environment step. We observe that in this setting it is hard to achieve competitive performance by directly learning the model online. To fix this problem, we propose *selective model learning*: for a batch $\mathcal{B}$ of data $(s, a, s')$ that are used for learning the model, we rank the data according to the model error function $\hat{\mathcal{E}}(s, h_{\mathrm{sml}})$, and use $x$ percent of the data that have small model errors to learn the model. By using this selective model learning approach, we hope to learn a partially accurate model that only focus on the dynamics which are easy to be learned. Our adaptive planning method can still benefit from such model since it is able to learn where the model has large errors and only adopt a small planning horizon at those states. We also note that using the model error function $\hat{\mathcal{E}}$ is different with directly computing the model error for each data in $\mathcal{B}$, since $\hat{\mathcal{E}}$ is learned by a reference policy and thus can provide more stable guidance for robust model learning. Another advantage of using $\hat{\mathcal{E}}$ is that we can tune the $h_{\mathrm{sml}}$ parameter, in which case we try to identify states whose nearby regions have large model error.

We denote AdaMVE with selective model learning by SML_AdaMVE. In all test domains, we use the *replay* reference policy and $H_{\max} = 3$ for both AdaMVE and Sml-AdaMVE. We tune $h_{\mathrm{sml}}$ from $\{1, 2\}$ and use $x = 50$ for selective model learning. For MVE, we tune the rollout horizon $H$ from $\{1, 3, 5\}$ and report the best result. Surprisingly, we find that $H = 1$ gives the best results in all test domains. We also compare our methods with STEVE, the state-of-the-art model-based value expansion method (Buckman et al., 2018). In our implementation of STEVE, we use 3 value functions, $H = 3$, and 3 independently online learned models to create ensembles. Results are presented in Fig 4 (f)-(i). With an online learned model, vanilla AdaMVE does not improve performance over the baselines, due to the difficulty to catch the error of an online updated model. However, by relaxing the model learning procedure using selective model learning, SML_AdaMVE outperforms all the baselines with an online learned model. Importantly, our proposed method has both the model and the model error as a single function, which is far less computationally expensive than the stochastic adaptive method STEVE, but show significantly better performance in practice.

## 6 CONCLUSION

We present a principled method to learn model errors by TD-learning in model-based reinforcement learning. Based on the learned model errors, an adaptive approach to select state-dependent planning horizons is introduced. Our proposed algorithm, AdaMVE, combines model-based and model-free reinforcement learning by adaptively selecting the rollout horizons in model-based value expansion. Empirical results shows that AdaMVE (i) successfully adapts the planning horizons according to the local correctness of the model, (ii) outperforms model-free, non-adaptive and stochastic-adaptive model-based baselines.

For the future work, we would like to combine our adaptive planning horizon method in other model-based RL approaches such as model predictive control and Monte Carlo tree search. Another future direction is how to learn a (partially correct) model online. In the experiments, we observe that simply fitting a neural network with minibatch and training L2 losses is not good enough and a better model learning method is needed. Our proposed selective model learning method is a preliminary attempt to solve this problem. We believe learning a reasonably good model online in complex high dimensional control tasks is an important problem and deserves thorough future studies.

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

# A  APPENDIX

## A.1  PSEUDOCODE

We provide the pseudocode for AdaMVE. We provide the pseudocode for discrete setting (based on DQN), and the model error is learned by the replay buffer reference policy. This code can be easily extend to continuous setting (based on DDPG) and model error learning with the other two reference policies as discussed in Section 3.1.

---

**Algorithm 1** Adaptive Mode-based Value Expansion

---

**Input:** maximum rollout horizon $H_{\max}$
Initialize the replay buffer $\mathcal{B}$ to capacity $N$
Initialize the approximate model $P_\nu$
Initialize action value function $Q_\theta$ (and/or the policy funtion $\pi_\theta$ if applicable)
Initialize state model error function $\hat{\mathcal{E}}_\phi$ with maximum rollout horizon $H_{\max}$
**for** $t = 0$ **to** $T - 1$ **do**
    Sample transitions using the $\epsilon$-greedy policy and store transitions in $\mathcal{B}$
    (If learn model) Sample transitions from $\mathcal{B}$ to learn the model $P_\nu$
    Sample transitions from $\mathcal{B}$ to learn the model error $\hat{\mathcal{E}}_\phi$
    Sample a batch of transitions $(s, a, r, s')$ from $\mathcal{B}$
    For each $s'$ in the batch get $\hat{\mathcal{E}}_\phi(s', h)$ for $h \in [0, H_{\max}]$
    For each $s'$ compute the multi-step value expansion target for each $h \in [0, H_{\max}]$
    Compute the target value $\tilde{V}^\pi_{\hat{P}, H_{\max}}(s')$ according to (9)
    Update $\theta$ using model free RL by using $\tilde{V}^\pi_{\hat{P}, H_{\max}}(s')$ as the target value
    Update target network parameters for both policy training and model error training, $\bar{\theta}$ and $\bar{\phi}$, with exponential decay
**end for**

---

## A.2 PROOF OF THEOREM 1

*Proof.* For any $0 \leq h \leq H$, define $U_h$ to be the $H$-step value expansion that rolls out the true model $P$ for the first $h$ steps and the approximate model $\hat{P}$ for the remaining $H - h$ steps:

$$U_h = \sum_{t=0}^{h-1} \gamma^t \mathbb{E}_{s_t \sim P_t^\pi(\cdot|s)} \left[ R^\pi(s_t) \right] + \sum_{t=h}^{H-1} \gamma^t \mathbb{E}_{s_t \sim \hat{P}_{t-h}^\pi \circ P_h^\pi(\cdot|s)} \left[ R^\pi(s_t) \right]$$
$$+ \gamma^H \mathbb{E}_{s_H \sim \hat{P}_{H-h}^\pi \circ P_h^\pi(\cdot|s)} \left[ \bar{V}(s_H) \right] , \tag{10}$$

where $\hat{P}_{t-h}^\pi \circ P_h^\pi(\cdot|s)$ denotes the distribution over states after rolling out $h$ steps with $P$ and $t - h$ steps with $\hat{P}$, i.e.

$$\hat{P}_{t-h}^\pi \circ P_h^\pi(\cdot|s) = \sum_{s' \in \mathcal{S}} P_h^\pi(s'|s) \hat{P}_{t-h}^\pi(\cdot|s') .$$

From the definition of $U_h$ we know $U_0 = \hat{V}_{\hat{P},H}^\pi(s)$ and $U_H = \hat{V}_{P,H}^\pi(s)$. Hence we have

$$\hat{V}_{\hat{P},H}^\pi(s) - \hat{V}_{P,H}^\pi(s) = U_0 - U_H = \sum_{h=0}^{H-1} (U_h - U_{h+1}) .$$

To analyze the difference between $U_h$ and $U_{h+1}$, we rearrange the terms in (10) in two different ways:

$$U_h = \sum_{t=0}^{h-1} \gamma^t \mathbb{E}_{s_t \sim P_t^\pi(\cdot|s)} \left[ R^\pi(s_t) \right] + \gamma^h \mathbb{E}_{s_h \sim P_h^\pi(\cdot|s)} \left[ \hat{V}_{\hat{P},H-h}^\pi(s_h) \right] , \tag{11}$$

$$U_h = \sum_{t=0}^{h} \gamma^t \mathbb{E}_{s_t \sim P_t^\pi(\cdot|s)} \left[ R^\pi(s_t) \right] + \gamma^{h+1} \mathbb{E}_{s_{h+1} \sim \hat{P}^\pi \circ P_h^\pi(\cdot|s)} \left[ \hat{V}_{\hat{P},H-h-1}^\pi(s_{h+1}) \right] . \tag{12}$$

Now applying (12) to $U_h$ and (11) to $U_{h+1}$, we can bound $U_h - U_{h+1}$ by

$$\sum_{t=0}^{h} \gamma^t \mathbb{E}_{s_t \sim P_t^\pi(\cdot|s)} \left[ R^\pi(s_t) \right] + \gamma^{h+1} \mathbb{E}_{s_{h+1} \sim \hat{P}^\pi \circ P_h^\pi(\cdot|s)} \left[ \hat{V}_{\hat{P},H-h-1}^\pi(s_{h+1}) \right]$$

$$- \sum_{t=0}^{h} \gamma^t \mathbb{E}_{s_t \sim P_t^\pi(\cdot|s)} \left[ R^\pi(s_t) \right] - \gamma^{h+1} \mathbb{E}_{s_{h+1} \sim P_{h+1}^\pi(\cdot|s)} \left[ \hat{V}_{\hat{P},H-h-1}^\pi(s_{h+1}) \right]$$

$$= \gamma^{h+1} \mathbb{E}_{s_h \sim P_h^\pi(\cdot|s), a_h \sim \pi(\cdot|s_h)} \left[ \mathbb{E}_{s' \sim \hat{P}(\cdot|s_h,a_h)} \left[ \hat{V}_{\hat{P},H-h-1}^\pi(s') \right] - \mathbb{E}_{s' \sim P(\cdot|s_h,a_h)} \left[ \hat{V}_{\hat{P},H-h-1}^\pi(s') \right] \right]$$

$$\leq \left\| \hat{V}_{\hat{P},H-h-1}^\pi \right\|_L \gamma^{h+1} \mathbb{E}_{s_h \sim P_h^\pi(\cdot|s), a_h \sim \pi(\cdot|s_h)} \left[ \sup_{\|f\|_L \leq 1} \mathbb{E}_{s' \sim \hat{P}(\cdot|s_h,a_h)} \left[ f(s') \right] - \mathbb{E}_{s' \sim P(\cdot|s_h,a_h)} \left[ f(s') \right] \right]$$

$$\leq K \gamma^{h+1} \mathbb{E}_{s_h \sim P_h^\pi(\cdot|s)} \left[ W^\pi(s_h) \right] . \tag{13}$$

The bound (13) also holds for the opposite direction $U_{h+1} - U_h$. Therefore

$$\left| \hat{V}_{\hat{P},H}^\pi(s) - \hat{V}_{P,H}^\pi(s) \right| \leq \sum_{h=0}^{H-1} |U_h - U_{h+1}| \leq K \sum_{h=0}^{H-1} \gamma^{h+1} \mathbb{E}_{s_h \sim P_h^\pi(\cdot|s)} \left[ W^\pi(s_h) \right] ,$$

which concludes the proof. $\square$

## A.3 EXTRA EXPERIMENTS ON FOURROOM

**Comparison with MVE using other planning horizon**. In Fig. 3 we compare AdaMVE with MVE using a fixed horizon $H = 5$. In this section we provide experiment results compared with MVE_$h1$ (fixed horizon $H = 1$) and MVE_$h3$ (fixed horizon $H = 3$). AdaMVE still applies $H_{\max} = 5$. AdaMVE clearly outperforms MVE-$h1$ and MVE-h3 on both domains in terms of sample efficiency and final performance.

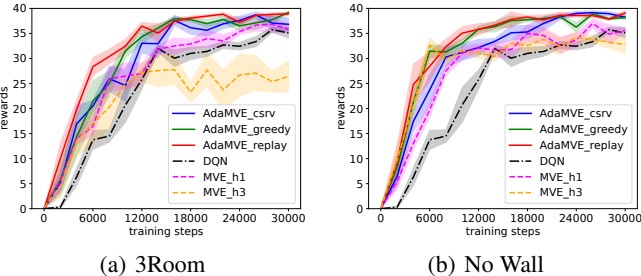

(a) 3Room            (b) No Wall

Figure 5: We compare AdaMVE with MVE_$h1$ (fixed planning horizon 1) and MVE_$h3$ (fixed planning horizon 3). AdaMVE outperforms both methods on both domains in terms of sample efficiency and final performance.

**Transferring Adaptivety to Different Tasks** If the environment dynamics is fixed, the model error learned in one task can be directly transferred to a different task. To illustrate this, we create a new task named FourRoom2, by changing the goal position in FourRoom from (15,15) to (2,18). In FourRoom2, we first train a model error function in FourRoom, then directly apply the pre-trained model error for AdaMVE. We include DQN and AdaMVE that learns from scratch in the new tasks as the baseline algorithms for comparison. Results are shown in Fig. 6. AdaMVE with the transferred model error is denoted by T-AdaMVE.

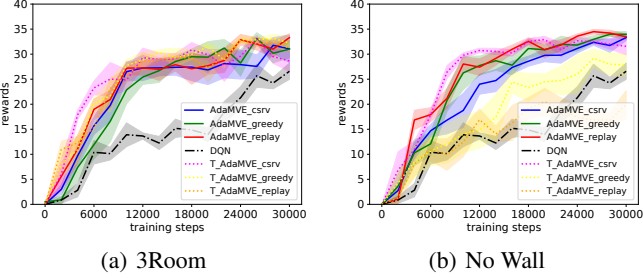

(a) 3Room            (b) No Wall

Figure 6: Evaluation of model error transformation.

Recall that the conservative policy tries to maximize the model error to provide a robust estimation. Hence, the model error learned by this policy in FourRoom should still be effective for FourRoom2. The result clearly confirms this intuition: with the pre-trained model error, AdaMVE_csrv learns much faster than the one learning from scratch. In comparison, the model error learned by the greedy policy and replay policy are less useful for transfer between the two tasks. When the no wall model is used, the transferred model error even brings negative effects. This is due to the fact that both the greedy policy and replay policy are based on the learning agent's behavior during training and therefore would change in different tasks.

## A.4 EXPERIMENT DETAILS

Table 1 lists the parameters used in gridworld experiments. Table 2 lists the parameters used in Mujoco Navigation. For selecting the mixing temperature $\tau$, in Gridworld we pick the best one (0.01) between 0.01 and 0.001. For Mujoco environments we use $\tau = 0.01$ without further tuning. In Swimmer, we use the double-Q trick to provide better performance (Fujimoto et al., 2018). For selective model learning, the reported results use $h_{\text{sml}} = 2$ for PointRoom, and $h_{\text{sml}} = 1$ for other domains.

Table 1: Parameters used in gridworld experiments.

| Parameter | Values |
|---|---|
| $\epsilon$-greedy | 0.2 |
| discount ($\gamma$) | 0.98 |
| $\tau$ | 0.01 |
| batch size | 128 |
| optimizer (all networks) | Adam |
| learning rate (q network) | 0.001 |
| learning rate (model error network) | 0.0001 |
| replay buffer initial size | 2000 |
| replay buffer size | $10^6$ |
| target network update interval | 1 |
| target network mixing coefficient | 0.001 |
| Q network for policy learning | (200, 200, 200) |
| Q network for model error | (200, 200, 200) |
| nonlinearity | ReLU |
| Maximum rollout steps $H_{\text{max}}$ | 5 |

Table 2: Parameters used in Mujoco Navigation.

| Parameter | Values |
|---|---|
| action noise for exploration | $\mathcal{N}(0, 0.1^2)$ |
| discount ($\gamma$) | 0.99 |
| $\tau$ | 0.01 |
| batch size | 128 |
| optimizer (all networks) | Adam |
| learning rate (all networks) | 0.0001 |
| A network weight decay (DDPG) | 1e-6 |
| Q network weight decay (DDPG) | 1e-3 |
| replay buffer initial size | 20000 |
| replay buffer size | $10^6$ |
| target netowrk update interval | 1 |
| target network mixing coefficient | 0.001 |
| Q network for both model error and policy learning | (300, 300, 300) |
| A network for both model error and policy learning | (300, 300) |
| nonlinearity | ReLU |
| Maximum rollout steps $H_{\text{max}}$ | 5 |

### A.5 SUPPORTING RESULTS

We provide supporting results in this section. Figure 7 (a) and (b) show the weighted average horizon $\bar{H}$ during training in PointRoom and PointMaze with a pretrained model. The maximum horizon $H_{\max}$ is set to 5. We can see that $\bar{H}$ over different states has very high variance, which is a sign of successful adaptation since the pretrained model is wrong at the states that are next to the wall (shorter rollout horizon) while being accurate at the states that are away from the wall (longer rollout horizon). PointMass has shorter overall rollout horizons than PointRoom because PointMaze has an extra wall and thus a smaller set of states where the pretrained model is accurate, hence enjoys less benefit of using AdaMVE over DDPG than in PointRoom.

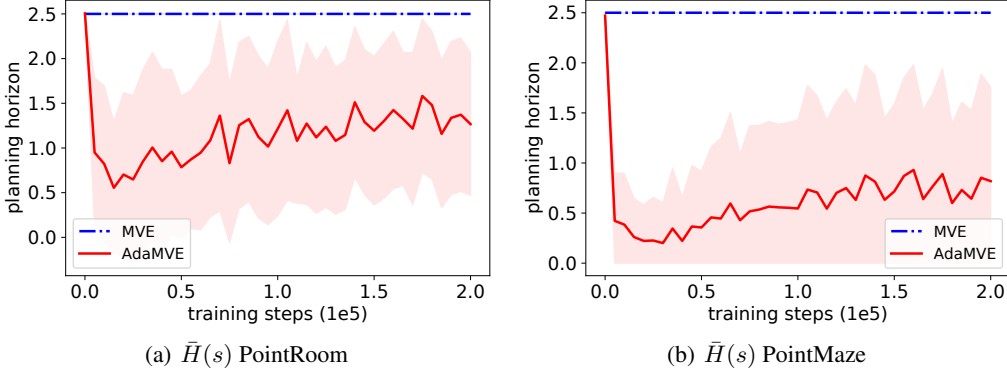

(a) $\bar{H}(s)$ PointRoom

(b) $\bar{H}(s)$ PointMaze

Figure 7: Supporting Results. When $H_{\max} = 5$, the maximum of weighted average horizon $\bar{H}$ is 2.5 according to the definition.

