# OpenReview forum: "Learning to Combat Compounding-Error in Model-Based Reinforcement Learning"
_ICLR.cc/2020/Conference — Reject_

### Official Review · AnonReviewer1 · 2019-10-19
**Official Blind Review #1**

**Rating:** 8

**Review:**

Learning to Combat Compounding-Error in Model-Based Reinforcement Learning

The authors study the problem of choosing the appropriate planning horizon in model-based reinforcement learning. We want to have a balance between the efficiency of planning into the future and while avoiding the compounding error brought by the learnt model.
And the authors propose to learn an error estimation model that is a function of current state, policy and horizon by estimating the compounding error as a new reward using policy evaluation.
Pros -
I think the novelty in this paper is most appealing for me to vote for a clear acceptance. The growth of model-based reinforcement learning is in a big need of a framework to work with the planning horizon.
The proposed method is a very elegant way of estimating the errors and choose the appropriate horizon.
Still, I think there is a lot of room for improvement, as summarized below.

Cons
- One thing that concerns me is that the maximum horizon H looks very small. A max horizon of 3, 5 or even 10 is still much smaller than the value we usually use in planning based reinforcement learning (MPC in your case). A common choice can be up to 30 or 50 for PETS based algorithms [1].

Does that indicate that the proposed framework is not stable or robust when the horizon is big? Or maybe that scale of horizon is not common in Steve and thus not tried in the experimental section?

- Experiments
For the high dimensional control tasks, only Half-Cheetah and Swimmer was tried. It will generalize the conclusion if the results of other benchmarking environments are tested, including complex tasks such as Humanoid, and easier tasks such as Cart-pole.

- Algorithms
The used baselines in the paper are generally not considered to be state-of-the-art. I guess it is sort-of fair comparison since you add the proposed component on top of the baselines. But extending the current state-of-the-art such as TD3 [2] / PETS

I also suggest adding this benchmarking paper into the related work, which empirically studies the compounding error in different state-of-the-art model-free and model-based reinforcement algorithms.

[1] Chua, K., Calandra, R., McAllister, R., & Levine, S. (2018). Deep reinforcement learning in a handful of trials using probabilistic dynamics models. In Advances in Neural Information Processing Systems (pp. 4754-4765).
[2] Fujimoto, Scott, Herke van Hoof, and David Meger. "Addressing function approximation error in actor-critic methods." arXiv preprint arXiv:1802.09477 (2018).
[3] Wang, Tingwu, Xuchan Bao, Ignasi Clavera, Jerrick Hoang, Yeming Wen, Eric Langlois, Shunshi Zhang, Guodong Zhang, Pieter Abbeel and Jimmy Ba. “Benchmarking Model-Based Reinforcement Learning.” ArXiv abs/1907.02057 (2019): n. pag.

**Experience Assessment:**

I have published in this field for several years.

**Review Assessment: Checking Correctness Of Derivations And Theory:**

I assessed the sensibility of the derivations and theory.

**Review Assessment: Checking Correctness Of Experiments:**

I assessed the sensibility of the experiments.

**Review Assessment: Thoroughness In Paper Reading:**

I read the paper thoroughly.

---

> ### Author Response · Authors · 2019-11-15
> **Response**
>
> We thank the reviewer for the valuable feedback. We are glad the reviewer found the problem studied to be interesting and important. Responses to the reviewer’s comments are addressed below. We are also happy to discuss further if the reviewer has additional concerns.
>
>
> “One thing that concerns me is that the maximum horizon H looks very small. A max horizon of 3, 5 or even 10 is still much smaller than … Or maybe that scale of horizon is not common in Steve and thus not tried in the experimental section? “
>
> -- As suggested in the review, the reason we only consider small scale of maximum horizon is that a large rollout horizon is not common in MVE based methods, e.g. STEVE. A short rollout horizon can be sufficient to utilize the benefit of a model when used in certain model-based deep RL algorithms. In previous works that combine value based RL methods with multi-step value expansion, they found that a small rollout horizon can be beneficial. For example, when applying DDPG+MVE in Mujoco continuous control tasks [1, 2, 3], a small rollout horizon can be effective in providing better performance than purely model-free algorithms. For DQN on discrete action domains (e.g. Atari), [4] tunes the number of steps used to get a multi-step return as a hyper-parameter. They found that among {1, 3, 5}, 3 works the best as the number of rollout steps. Although the maximum horizons in these domains (Mujoco, Atari) are usually quite large (>~500), the observations in these previous works support that a small rollout horizon (1-5) can be helpful on these domains.
>
> A larger max-horizon may be needed for other model-based RL algorithms such as MPC based methods, e.g. PETS. As we mentioned in our conclusion section, we would leave combining our adaptive planning horizon method with other model-based RL approaches as future work. We note that our method considers the problem of learning accumulative model errors as policy evaluation in finite-horizon MDP, and it is valid for any large maximum planning horizon. For model-based algorithms that require large planning horizons, our method for estimating model errors and adapting planning horizon can be directly applied in those cases.
>
>
> “For the high dimensional control tasks, only Half-Cheetah and Swimmer was tried. It will generalize the conclusion if the results of other benchmarking environments are tested, including complex tasks such as Humanoid, and easier tasks such as Cart-pole.”
>
> -- We agree that testing on more tasks will certainly make our results stronger. In the online model learning setting as considered in our experiments, it is very difficult to learn a partially accurate model in complex domains (such as Humanoid) by simply fitting a neural network with one minibatch per environment step.  For a poorly learned model where the model error is large almost everywhere, our adaptive horizon selection strategy (equation 8) assigns negligible weights to horizons above zero, which makes our algorithm behaves similarly as the model-free baseline. To get a good model, we may need a much larger neural network with many more gradient steps per environment step (as in [3]), which requires a significant amount of computation resource.
>
>
>
> “The used baselines in the paper are generally not considered to be state-of-the-art. I guess it is sort-of fair comparison since you add the proposed component on top of the baselines. But extending the current state-of-the-art such as TD3 / PETS”
>
> We apply DQN and DDPG as the model-free baselines in our experiments, as in most of our test domains, these algorithms already have reasonably good performance (e.g. TD3 will only give a marginal improvement over DDPG on those tasks). Also, as suggested in the review, we believe this choice of the baseline is fair since we add the proposed component on the top of the baselines. Furthermore, we do extend the current state-of-the-art version in domains where DDPG performs not well. For example, in Swimmer we use the double-Q-function trick proposed by TD3 in DDPG to get a good model-free baseline performance. This is mentioned in Appendix A.4.
>
>
> "I also suggest adding this benchmarking paper into the related work, which empirically studies the compounding error in different state-of-the-art model-free and model-based reinforcement algorithms."
>
> Thanks for your suggestion. We have included this paper in the related work in the updated version.
>
>
> [1] Feinberg, V., Wan, A., Stoica, I., Jordan, M.I., Gonzalez, J.E. and Levine, S., 2018. Model-based value estimation for efficient model-free reinforcement learning.
>
> [2] Buckman, J., Hafner, D., Tucker, G., Brevdo, E. and Lee, H., 2018. Sample-efficient reinforcement learning with stochastic ensemble value expansion.
>
> [3] Janner, M., Fu, J., Zhang, M. and Levine, S., 2019. When to Trust Your Model: Model-Based Policy Optimization.
>
> [4] Hessel, Matteo, et al. Rainbow: Combining improvements in deep reinforcement learning.

---

### Official Review · AnonReviewer3 · 2019-10-27
**Official Blind Review #3**

**Rating:** 1

**Review:**

#rebuttal responses

The authors' reply does not convince me, and I still think the paper has some problems:
(1) I do not believe that the cumulative model-error can not be learned efficiently;
(2) Experimental results are weak as some baselines do not converge!

Thus I keep my rating as reject.

#review
This paper proposes a new adaptive model-based value-expansion method, AdaMVE, that decides the planning horizon of the learned model by learning the model-error. The model-error is learned by temporal difference methods.
Experimental results show that AdaMVE beats MVE, STEVE, and DDPG in several environments.

Overall the paper is well written.  The paper proposes an interesting question: how to adaptively change the planning horizon based on the state-dependent model-error? Firstly, The authors upper bound the cumulative target error by the cumulative model-error.  Then the cumulative model-error is learned by the temporal difference method. With the learned cumulative model-error function over different rollout steps, the planning horizon is decided by a softmax policy.

However, I do not think that learning an upper bound of the target error helps to determine the value of H, as there is no justification that the gap between the target error and the model error is small theoretically.  I also doubt that the cumulative model-error can be learned without large loss, as there are no plots of W in this paper.

The authors claim that it is expensive to retrain the model error for the current policy at every step, thus they use some reference policy. I think it is ok, but I want to see the results of AdaMVE using the updated current policy, or updating the Q function before improving the policy. Adding a figure showing the change of H in the training helps to motivate this paper.

Finally, AdaMVE is only compared in two MuJoCo environments. Baselines in other environments are only trained in 1e5 steps, thus the experimental results are not convincing.

I am happy to change my opinion on this paper if authors give better motivation and the detail of the learning.

**Experience Assessment:**

I have published one or two papers in this area.

**Review Assessment: Checking Correctness Of Derivations And Theory:**

I assessed the sensibility of the derivations and theory.

**Review Assessment: Checking Correctness Of Experiments:**

I carefully checked the experiments.

**Review Assessment: Thoroughness In Paper Reading:**

I read the paper thoroughly.

---

> ### Author Response · Authors · 2019-11-15
> **We thank the reviewer for valuable feedback. Responses to the reviewer’s comments are addressed below.**
>
> “However, I do not think that learning an upper bound ... the model error is small theoretically.”
>
> -- Dealing with model compounding error is an important issue in model-based RL when the model is inaccurate. However, the model compounding error cannot be computed directly, which motivates our work to derive a learnable upper bound (the cumulative model error) on it (Thm 1).
>
> Once the cumulative model error is learned, the remaining question is to decide h given upper bound estimates of the target error for each h. The softmax policy (Eq 8)  aims at selecting h where the cumulative model error is small while keeping as many such h (with small error) as possible to utilize longer horizons.
>
> The reviewer worries that selecting h according to the upper bound might be problematic when the upper bound is loose. However, if the upper bound is loose (the model error is high while target error is small), our adaptive strategy (Eq 9) becomes conservative and will not bring any negative effect on policy learning. This is because in Eq 9 we also consider h=0 that has zero model error, which makes any h with large error get a small weight, and the expanded value reduced to the model-free target. However, the fact that our method outperforms model-free baselines and that the selected h is not close to 0 (see Fig 3 & Fig 7 in A.5) is clear evidence to support that our upper bound is *not* loose.
>
>
>
> “I also doubt that the cumulative model-error can be learned without large loss.”
>
> -- The cumulative model-error can be learned by minimizing the TD error (Eq 7). In our scenario, W can be directly computed for any (s,a,s’) using the l2 norm (see Sec 2.2 and 3.1). By this formulation, learning the cumulative model-error is just another value learning problem in MDP and is no harder than learning the Q values. (And probably easier, as the cumulative model-error is of finite horizon.) Like in Q-learning (DQN, DDPG), it is hard to tell whether the Q-values are well learned or not by simply plotting the training loss (what values are *large* and what values are *small*). It is more important that the learned values are good enough for its purpose: In Q-learning whether the learned values induce a good policy, in our case whether the learned model error is effective in deciding planning horizons that improve the performance. (E.g., in DQN and DDPG, people have observed that there is always a gap between learned and real Q values, but the learned values can be good enough to induce a good policy.)
>
>
>
> “I want to see the results of AdaMVE using the updated current policy or updating the Q function before improving the policy.”
>
> -- One of our reference policies, the *greedy* reference policy is the current policy. (see Sec 3.1 & Fig3 (g)-(i)).
>
>
> “Adding a figure showing the change of H in training helps to motivate this paper.“
>
> -- We include two supporting figures showing the change of the weighted average horizon during training in mujoco navigation using a pre-trained model. (See Fig 7 in Appendix A.5 in our updated version). We can see the weighted average horizon over different states has a high variance, which is a sign of successful adaptation.
>
>
> “Finally, AdaMVE is only compared in two MuJoCo environments.”
>
>
> -- We agree that testing on more tasks will certainly make our results stronger. However, our main contribution is to propose a novel approach to leverage an inaccurate model in model-based RL, rather than pursuing state-of-the-art performance on hard tasks. To demonstrate the effectiveness of our method, we choose a diverse set of tasks:  simple gridworlds, Mujoco navigation, Mujoco locomotion. The advantage of selecting this set of tasks instead of a set of harder but similar ones (e.g. a full set of Mujoco locomotion tasks) is that some of these tasks allow more interpretable demonstrations. E.g., the gridworld allows us to visualize the adaptive horizon over all states. In Mujoco navigation, the flexibility in customizing the walls allows us to pretrain a partially correct model with a specific meaning: by removing the walls the pretrained model is correct with locomotion but inaccurate with interaction to the wall. This sets a clear expectation that a successful adaptive strategy should be able to utilize this pretrained model. With these explanations we hope we can convince the reviewer that we put a sufficient amount of effort in demonstrating the effectiveness of our method in various ways so that evaluation on more Mujoco locomotion tasks would be a bonus rather than a necessity.
>
>
> “Baselines in other environments are only trained in 1e5 steps”
>
> -- On these tasks, the best performing algorithm has converged to near optimal performance given the training budgets (2e5 steps). We also want to note that the 2e5 steps is a reasonable choice in Mujoco navigation tasks and it is also adopted in previous works [1].
>
> [1] Wu, Y., et al. The laplacian in rl: Learning representations with efficient approximations.

---

### Official Review · AnonReviewer2 · 2019-10-31
**Official Blind Review #2**

**Rating:** 6

**Review:**


# Rebutal Respons:

Planning Horizon:
- I agree small horizons can speed up learning. However, if we want to drastically reduce the sample complexity, we need longer planning horizons. Therefore, we should be looking further in this direction but this is out of scope for this paper.

Model Learning:
- Good idea to update the conclusion and treat the online model learning as future work.

Figure 3:
You could plot axes a - f in a single row and remove the colorbar for each individual plot. Just do one colorbar for all plots. This would also improve the comparability of the different methods. Currently, the color-coding is different for each axis, which is bad practice. Axes g - i can be reshaped in a separate figure. Furthermore, figure 2 is not necessary as the 4 room environment is depicted in Fig. 3 a - f and and one could reference these axes.

=> I keep my rating as weak accept.

# Review:
Summary:
The paper proposes an adaptive scheme to adapt the horizon of the value function update. Therefore, the sample efficiency should be increased and the value function should be learned faster.

Conclusion:
The problem of learning good policies from partially correct models is very interesting and important. The proposed approach is technically sound and reasonable. The experiments highlight the qualitative as well as quantitative performance. Furthermore, the quantitative performance is compared to state-of-the-art methods. I cannot comment on the related work as I am not familiar with the baselines MVE & STEVE.

My Main Concerns are:

- The roll-out horizon is 3-5 timesteps. This horizon is really short especially for problems with strong non-linearities and high sampling frequencies. For such systems 5 timesteps would only correspond to 0.5s (10Hz) or 0.05s (100Hz) and I am uncertain whether these short horizons really help for such problems.

- The specialized online model learning algorithm seems quite hacky. It feels like it was introduced last minute to make it work. The overall question of how should we learn a model optimally is super important and should be addressed within a separate paper (and more thoroughly). I would even propose to remove the online learning section from this paper as it is just too hacky and without relation to prior work or context.

- Could the authors please update figure 3 as the figure has too much whitespace. This whitespace could be used to enlarge the individual axis when the axis are rearranged.


**Experience Assessment:**

I have read many papers in this area.

**Review Assessment: Checking Correctness Of Derivations And Theory:**

I assessed the sensibility of the derivations and theory.

**Review Assessment: Checking Correctness Of Experiments:**

I assessed the sensibility of the experiments.

**Review Assessment: Thoroughness In Paper Reading:**

I read the paper at least twice and used my best judgement in assessing the paper.

---

> ### Author Response · Authors · 2019-11-15
> **Response to Reviewer #2**
>
> We thank the reviewer for the valuable feedback. We are glad the reviewer found the problem studied to be interesting and important. Responses to the reviewer’s comments are addressed below. We are also happy to discuss further if the reviewer has additional concerns.
>
>
> “The roll-out horizon is 3-5 timesteps... I am uncertain whether these short horizons really help for such problems.”
>
> -- A short rollout horizon can be sufficient to utilize the benefit of a model when used in certain model-based deep RL algorithms, as shown in previous works. In our experiments, the learning algorithms are value-based deep RL algorithms (DQN, DDPG) combined with multistep model-based value expansion (MVE).  In previous works that combine value-based RL methods with multi-step value expansion, they found that a small roll-out horizon can be beneficial. For example, when applying DDPG+MVE in Mujoco continuous control tasks [1, 2, 3], a short rollout horizon can be effective in providing better performance than purely model-free algorithms. For DQN on discrete action domains (e.g., Atari), [4] tunes the number of steps used to get a multi-step return as a hyper-parameter. They found that among {1, 3, 5}, 3 steps of rollout works the best. Although the maximum horizons in these domains (Mujoco, Atari) are usually quite large (>=500), the observations in these previous works support that a short rollout horizon (1-5) can be helpful on these domains.
>
> Indeed a larger max-horizon may be needed for other model-based RL algorithms. We note that our method considers the problem of learning accumulative model errors as policy evaluation in the finite-horizon MDP, and it is valid for any large maximum planning horizon. For model-based algorithms that require large planning horizons, our method for estimating model errors and adapting the planning horizon can be directly applied in those cases.
>
>
>
> “The specialized online model learning algorithm seems quite hacky... The overall question of how should we learn a model optimally is super important and should be addressed within a separate paper (and more thoroughly)”
>
> -- We agree that how to learn a (maybe partially correct) model online is an important problem that is worth a separate and thorough study. We present these results because of the following. (i) We (and potential readers, according to some feedback we got) are curious about whether our method can work with an online-trained model. (ii) We found that a better online model-learning method does help with performance. These results can serve as supporting evidence for that simply fitting a neural network with minibatch and training L2 losses is not good enough and a better model learning method is needed. Our proposed selective model learning method is a preliminary attempt in this direction.  We have updated the conclusion section of our paper to clearly state that our online model learning method is a preliminary attempt and it is an important problem that needs future study.
>
>
>
> “Figure 3”
>
> -- We have not figured out how to rearrange the space nicely. We will try to do it in the final version. We would also appreciate any suggestions from the reviewer.
>
>
>
> [1] Feinberg, V., Wan, A., Stoica, I., Jordan, M.I., Gonzalez, J.E. and Levine, S., 2018. Model-based value estimation for efficient model-free reinforcement learning.
>
> [2] Buckman, J., Hafner, D., Tucker, G., Brevdo, E. and Lee, H., 2018. Sample-efficient reinforcement learning with stochastic ensemble value expansion. In Advances in Neural Information Processing Systems.
>
> [3] Janner, M., Fu, J., Zhang, M. and Levine, S., 2019. When to Trust Your Model: Model-Based Policy Optimization.
>
> [4] Hessel, Matteo, et al. "Rainbow: Combining improvements in deep reinforcement learning." Thirty-Second AAAI Conference on Artificial Intelligence. 2018.

---

### Decision · Program_Chairs · 2019-12-19

**Decision:**

Reject

**Comment:**

The paper received mixed reviews: R (R3), WA (R2), A (R1). AC has read the reviews, rebuttal and paper. AC is concerned about the short planning horizon, which seems like a major issue: (i) as R1 notes, most MPC algorithms use much longer horizons as they find it helps performance and (ii) the claim of the approach to be able to pick the planning horizon is moot if its dynamic range is small.  Overall, the paper is very borderline. The idea is interesting but without addressing longer horizons, the contribution is limited. Under guidance from the PCs, the AC feels that the paper just falls below the acceptance threshold and thus cannot be accepted unfortunately. The work is definitely interesting however and should be revised for a future submission.